# Squalene-Based Nano-Assemblies Improve the Pro-Autophagic Activity of Trehalose

**DOI:** 10.3390/pharmaceutics14040862

**Published:** 2022-04-14

**Authors:** Giulia Frapporti, Eleonora Colombo, Hazem Ahmed, Giulia Assoni, Laura Polito, Pietro Randazzo, Daniela Arosio, Pierfausto Seneci, Giovanni Piccoli

**Affiliations:** 1Department of Cellular, Computational and Integrative Biology (CIBIO), Via Sommarive 9, Povo, I-38123 Trento, Italy; giulia.frapporti@unitn.it (G.F.); giulia.assoni@unitn.it (G.A.); 2Chemistry Department, Università Statale di Milano, Via Golgi 19, I-20133 Milan, Italy; eleonora.colombo@unimi.it (E.C.); hazem.ahmed@iit.it (H.A.); 3Istituto Italiano di Tecnologia (IIT), Via Morego 30, I-16163 Genova, Italy; 4Istituto di Scienze e Tecnologie Chimiche (SCITEC) “Giulio Natta”, Consiglio Nazionale delle Ricerche (CNR), Via G. Fantoli 16/15, I-20138 Milan, Italy; laura.polito@scitec.cnr.it; 5Promidis srl, Via Olgettina 60, I-20132 Milan, Italy; p.randazzo@promidis.it; 6Istituto di Scienze e Tecnologie Chimiche (SCITEC) “Giulio Natta”, Consiglio Nazionale delle Ricerche (CNR), Via Golgi 19, I-20133 Milan, Italy; daniela.arosio@scitec.cnr.it

**Keywords:** autophagy inducers, trehalose, squalene conjugates, nanoassemblies, reduction-labile linkers, cancer, neurodegeneration

## Abstract

The disaccharide trehalose is a well-established autophagy inducer, but its therapeutic application is severely hampered by its low potency and poor pharmacokinetic profile. Thus, we targeted the rational design and synthesis of trehalose-based small molecules and nano objects to overcome such issues. Among several rationally designed trehalose-centered putative autophagy inducers, we coupled trehalose via suitable spacers with known self-assembly inducer squalene to yield two nanolipid-trehalose conjugates. Squalene is known for its propensity, once linked to a bioactive compound, to assemble in aqueous media in controlled conditions, internalizing its payload and forming nanoassemblies with better pharmacokinetics. We assembled squalene conjugates to produce the corresponding nanoassemblies, characterized by a hydrodynamic diameter of 188 and 184 nm and a high stability in aqueous media as demonstrated by the measured Z-potential. Moreover, the nanoassemblies were characterized for their toxicity and capability to induce autophagy in vitro.

## 1. Introduction

Autophagy is a self-digestion process that allows the lysosomal degradation of misfolded protein aggregates and damaged organelles. Clearance of proteins plays a pivotal role in neuronal homeostasis; compromised protein clearance hampers central nervous system function and eventually leads to clinical manifestations [1,2]. Studies in complementary animal and cellular models suggest that autophagy may rescue from the pathological impact of misfolded protein aggregation in neurodegenerative disorders, including Alzheimer’s [3]. and Parkinson’s disease [4]. Autophagy is instrumental to guarantee cellular integrity and viability, balancing the cell’s energy consumption and maintaining homeostasis [5,6]. In basal conditions, autophagic activity is low. Upon stress, autophagy upregulation will in turn boost cellular degradation, thus supporting the recycling of amino acids to assure the cellular integrity. Autophagy is known to interact with and control canonical apoptosis. As such, the role of autophagy in cancer has been extensively studied; autophagy deregulation is now considered a key feature for tumor progression [7]. Numerous tumor suppressor oncogenes have been identified as autophagic regulators, including the oncosuppressor p53, which may halt or induce autophagy depending on its cellular localization and the underlying signaling pathways [8]. Accumulating evidence shows that downregulating key autophagic genes correlates with the promotion of tumorigenesis in some solid tumors such as breast, ovarian and prostate cancers [9]. 

Therefore, the pharmacological induction of autophagy may be an effective therapeutic strategy not only in several neurodegenerative disorders, including PD and AD [10], but also in cancer [11]. 

Trehalose is a naturally occurring disaccharide formed by an α,α-1,1-glucosidic bond between two α-glucose units. It is biosynthesized mostly by anhydrobiots, which include bacteria, yeast, nematodes, rotifers, tardigrades, certain crustaceans and insects [12]. Trehalose plays a pivotal role in various types of stress-tolerance, including radiation, cold and dehydration stress in these creatures [13,14,15]. Trehalose activates autophagy in vitro and in vivo [16]. Two main mechanisms have been proposed to explain the biological activity of trehalose. Namely, it blocks glucose transporters and thus triggers induced AMPK-dependent autophagy [17], or it acts by the lysosomal activation of the transcription factor TFEB, a master regulator of the autophagy-lysosome pathway function [18]. Recently, we found that trehalose can rescue protein aggregation and motor impairment in a rodent model of PD [4]. However, the pharmacokinetics properties of trehalose are poor. Previously, we characterized the pro-autophagic properties of gold nanoparticles functionalized with trehalose [19]. Here, we describe the synthesis and the biological activity of two squalene-trehalose constructs and of the corresponding nanoassemblies (NAs). More in detail, we characterized their toxicity and their efficacy in terms of autophagy induction in an in vitro model of cervical carcinoma.

## 2. Results

### 2.1. Synthesis of Mono- and Bis-Squalene-Trehalose Conjugates

In order to obtain either a mono- (target compound **1a**—Sq-mono, 1:1 squalene-trehalose conjugate) or a bis-squalenylated trehalose construct (target compound **1b**—Sq-bis, 2:1 squalene-trehalose conjugate), we envisaged the synthetic strategy described in Figure 1.

Namely, squalene-dithiolinker carboxylate **3** [20] was coupled in a 1:1 ratio with TMS-protected trehalose **2** [21] (step a, Figure 1). Both TMS-protected mono-squalenylated **4a** and bis-squalenylated trehalose **4b** were isolated in poor, unoptimized yields after chromatography. Both intermediates **4a** and **4b** were deprotected with AcOH in refluxing MeOH (step b, Figure 1), yielding pure 6-squalene-trehalose conjugate **1a** and 6,6′-bis- squalene-trehalose conjugate **1b** as self-assembling constructs.

### 2.2. Assembly of Mono- and Bis-Squalene-Trehalose NAs

The assembly of both NAs was carried out using a standardized method [20]. Namely, squalene-trehalose conjugates **1a** and **1b** were dissolved in THF (step a, Figure 2) and then added dropwise to stirred MilliQ water, reaching a ≈2 mg/mL concentration (step b). After stirring for 5 min, THF was thoroughly evaporated (step c, Figure 2).

Both NAs **5a** and **5b** were obtained as opalescent solutions and were structurally characterized in terms of hydrodynamic diameter (HD) and Z-potential before submitting them to biology profiling as putative autophagy inducers. Their features are shown in Table 1.

Dynamic light scattering (DLS) confirmed the formation of NAs in aqueous media. Their low polydispersity index values (PI < 0.2) confirmed the mono-dispersion of the colloidal solution of each NA, while their strongly negative Z-potential (<−35 mV) suggested a high electrostatic repulsion, i.e., good colloidal stability. Finally, hydrodynamic diameters were satisfyingly low for squalene-based NAs (<200 nm).

### 2.3. Biological Characterization of Trehalose-Based Small Molecules and NAs: Toxicity

First, we profiled the cytotoxicity and autophagy induction of our conjugates and NAs in a cellular model. In particular, we treated HeLa cultures for 24 h at 37 °C with either the two NAs **5a** and **5b** (at estimated adjusted concentrations of free trehalose in water equal to 10, 25 and 50 μM), their non-assembled squalene-trehalose precursors **1a** and **1b** (10, 25 and 50 μM) and each individual component of such constructs, namely 100 mM trehalose in water, 50 μM squalene, 50 μM 4,4′-dithiodibutyric acid (hereinafter linker) and relative vehicle (DMSO). At first, we determined the in vitro toxicity of each sample via the MTT assay. Upon 24 h treatment, **1b** and its related NA **5b** elicited a dose-dependent toxicity at concentrations higher than 10 μM (Figure 1A). In particular, both the non-assembled bis-squalene-trehalose precursor **1b** and its related NA **5b** showed significant cytotoxicity at 25 μM (≈65% viable cells with both **1b** and NA **5b**) and 50 μM (45% viable cells with **1b**, ≈25% viable cells with NA **5b**). Rather, mono-squalene-trehalose precursor **1a** and its related NA **5a** were not overtly toxic at 10 and 25 μM and triggered only minor toxicity at 50 μM (Figure 1B, ≈75% viable cells with both **1a** and **5a**).

### 2.4. Biological Characterization of Trehalose-Based Small Molecules and NAs: Efficacy

Next, we assessed whether the NAs, their non-assembled precursors and individual components induced autophagy upon 24 h of treatment. In particular, we tracked the mobility shift from LC3BI to LC3BII, which is a proxy for the induction of autophagy as well as the amount of LC3BII, itself related to the number of autophagosomes. α-Tubulin was used as an internal control in the assays. Squalene-trehalose precursor **1a** did not activate autophagy at tested concentrations (10 and 25 μM, Figure 2A and Figure 3A). Conversely, **1b** and both NAs efficiently enhanced autophagosome formation even at 10 μM, as suggested by the relative amount of LC3BII (Figure 2A–C and Figure 3A). Considering the autophagic flux, NA **5b** was already active at 10 μM, while NA **5a** showed positive effects at ≥25 μM concentration (respectively, Figure 2B,C and Figure 3B and Appendix A).

The increase in LC3BII indicates the accumulation of autophagosomes and does not guarantee their autophagic degradation. However, if the amount of LC3BII further accumulates in presence of a lysosomal protease inhibitor, this would indicate the promotion of autophagic flux. Therefore, we analyzed the biological effect of trehalose (25 μM), NAs **5a** (25 μM) and **5b** (10 μM) in presence of bafilomycin (100 nM), a well-known lysosomal blocker [22]. In detail, we added bafilomycin during the last 2 h of the treatment with NAs. Noteworthy, we noticed that LC3BII further accumulated upon co-administration of **5b** and bafilomycin (Figure 4A,B). In a complementary approach, we took advantage of a tandem fluorescent reporter GFP-mCherry-LC3B. Given the low pH of the autolysosome, the green fluorescence from the acid-sensitive GFP is lost upon fusion of the autophagosome with the lysosome. Conversely, the red fluorescence emitted by the acid-insensitive mCherry is not lost until the proteins are completely degraded. Such a double tag strategy allowed us to discriminate autophagosomes from autolysosomes [23,24]. 

Thus, we treated HeLa cells expressing the GFP-mCherry-LC3B reporter for 24 h with trehalose (positive control) and both NAs at the highest concentration lacking toxic effect, i.e., 10 μM for **5b** and 25 μM **5a**. We noticed that both **5b** and **5a** induced a significant increase of the number for both autophagosomes and autophagolysosomes (Figure 5A,B). The treatment did not alter the ratio between autophagosomes and autophagolysosomes, suggesting that it did not cause any gross alteration of the physiological autophagic flux (Figure 5C).

As our constructs and NAs included a disulfide bond connecting the squalene carrier and trehalose, we reasoned that a reducing agent may allow the release of functionalized trehalose through a prodrug-like mechanism. Thus, we investigated whether any intracellular reducing agent may play a role in the biological effect of the NAs.

Glutathione (GSH) is the major component of cellular antioxidant systems. Buthionine sulfoximine (BSO) depletes cellular GSH by inhibiting γ-glutamylcysteine synthetase (Figure 6A), a key enzyme in the GSH synthesis pathway [25]. We treated HeLa cells for 24 h with **5b** (10 μM) and **5a** (25 μM) either alone or in presence of 1 mM BSO. The treatment with BSO neither did affect significantly cell viability *per se*, nor modified the toxicity profile of our NAs (Figure 6B). Noteworthy, 1 mM BSO significantly impaired the effect of our NAs on the autophagic process by preventing the increase on autophagosome formation induced by 10 μM **5b** or 25 μM **5a** (Figure 6C and Figure 7A,B).

Altogether, these observations suggest that NAs **5a** and **5b** induced autophagy upon intracellular release of trehalose.

## 3. Discussion

Trehalose is a powerful autophagic inducer that holds promising therapeutic opportunities.

A major obstacle against its application as a therapeutic agent is its poor permeability across biological membranes. In fact, trehalose is highly hydrophilic, thus hampering its passive cell permeation [15]. Mammalian cells do not express trehalose transporters in their membranes. Moreover, the trehalose catabolic enzyme, trehalase, is highly expressed in the gut, causing trehalose hydrolysis to glucose upon *per os* administration [26]. Thus, trehalose-based treatment in humans is cumbersome and requires high dosages. For example, a clinical trial to test the efficacy of trehalose in treating oculopharyngeal muscular dystrophy entailed trehalose administration via a 1 h IV infusion at a 0.75 g/kg dosage (ClinicalTrials.gov Identifier: NCT04226924, accession date: 14 April 2022).

Nano lipid-drug conjugates couple a drug to biocompatible lipids and may represent a powerful tool to ameliorate the pharmacokinetics and improve the therapeutic index of the original drugs [27]. In particular, nano-assembled squalene-based conjugates have been successfully exploited in several therapeutic applications [28]. In water, squalene-based conjugates spontaneously assemble into NAs which enclose the bioactive molecule. NAs are typically internalized by the cells through clathrin-dependent and -independent endocytic pathways [29]. 

Squalene-based NAs are an attractive strategy to achieve intracellular delivery of trehalose, thus overcoming its poor permeability profile and possibly reducing its dosage in preclinical and clinical testing. However, our previous attempt to improve trehalose pharmacological activity via a squalene-based NA mostly failed [21]. Previously, we employed NAs based on covalent bound squalene-trehalose conjugates. As such, upon internalization, those NAs disassembled and released the unmodified construct, i.e., trehalose covalently bound to squalene. We reasoned that such a bulky squalene structure may significantly hamper the biological activity of trehalose.

Here, instead, we chose to conjugate squalene and trehalose via a biologically labile linker. In particular, we coupled two chemical entities via a biolabile disulfide bond.

In eukaryotic cells, disulfide bond stability tightly depends on the pH. In physiological conditions, disulfide bond formation happens principally in the oxidizing environment of the endoplasmic reticulum. Conversely, sulfhydryl groups of cysteine residues (Cys-SH) remain reduced at physiological pH in the reducing environment of the cytoplasm. Thus, the cytoplasm disfavors the existence of disulfide bonds. Accordingly, we designed the disulfide bond-containing NAs here described to facilitate the release of trehalose. Upon cellular internalization of NAs and subsequent disulfide bond reduction, trehalose is released with one (**5a**) or two (**5b**) short alkylthiol chains, which should eventually be aspecifically cleaved in the cytoplasm by esterases, releasing free trehalose.

Trehalose-containing NAs demonstrated a higher efficacy than free trehalose in terms of autophagy induction. Such an effect was dependent on the intracellular reduction of the disulfide bond, after NA internalization. Upon reduction of cellular GSH, we noticed a significant decline of the biological activity of our NAs. Notwithstanding their different composition and molecular weight, **5a** and **5b** showed a similar hydrodynamic diameter. Interestingly, **5b** demonstrated a higher effect on autophagy. The lack of an overt effect of **5a** once co-administered with bafilomycin may be due to the specific experimental conditions, as we may need to optimize timing and compound concentration. Conversely, we believe that the two residues of squalene present in compound **5b** self-assemble in order to display on the outer surface the trehalose residue. Therefore, the resulting bis-squalenylated nanoassemblies **5b** have a more hydrophobic core, do not increase their hydrodynamic diameter in respect to monosqualenylated **5a**, and may have higher permeability through the cellular membrane.

Finally, we noticed that our NAs induced autophagy at a low concentration and showed cytotoxicity at a higher concentration in HeLa cells, a well-established cellular model of cervical carcinoma. Indeed, our NAs need GSH to release trehalose, and this may stress such cells; however, pharmacological reduction of GSH did not overtly hamper cell viability. The transient release of trehalose with one (**5a**) or two (**5b**) short-lived alkylthiol chains may contribute to the observed cytotoxicity of such NAs. Conversely, autophagy is involved in several mechanisms, controlling cellular viability and integrity. Therefore, in our in vitro model, autophagic induction may precede cancer cell death. Indeed, autophagy may act as a tumor suppressor in the initial stages by clearing any potentially harmful organelle and protein and, eventually, may induce cancer cell death [11,30]. However, in the advanced stages of cancer, autophagy may help cancer cells to handle stressful conditions and assist them in their survival, as reviewed in [31]. Clearly, a full understanding of the potential of autophagy as a therapeutic target in cancer requires further studies.

## 4. Conclusions

Current data support the hypothesis that autophagy modulation could arise as a new therapeutic avenue for the treatment of multiple disorders, including cancer and neurodegenerative diseases. While bona fide autophagy modulators are not yet included in clinical practice, on-going investigations will lead to the development of safe autophagy modulators to be administered to patients.

In particular, squalene-based NA **5b** has been shown to improve trehalose pharmacological properties, but further investigation is needed to refine our knowledge of its properties and improve our NA lead for in vivo testing.

## 5. Materials and Methods

### 5.1. Synthesis

#### 5.1.1. General

Oven-dried glassware was used to carry out chemical reactions, and dry solvents under nitrogen atmosphere were employed. Solvents were purchased from Sigma Aldrich and used as such. Chemical reagents were also purchased from Sigma Aldrich and checked for integrity before using them. Direct phase flash chromatography columns were run with silica gel (240–400 mesh, Merck, Darmstadt, Germany). Reaction monitoring by thin layer chromatography (TLC) entailed Merck-precoated 60F_254_ plates. Reactions were monitored by TLC on silica gel, using as a direct detection method UV light at 254 nm, or by charring either with a 50% H_2_SO_4_ or with a 1% permanganate solution. ^1^H-NMR and ^13^C-NMR spectra were recorded in either CDCl_3_, CD_3_OD or DMSO-d6, depending on compounds’ solubility, on Bruker DRX-400 and Bruker DRX-300 instruments. Chemical shifts (δ) for proton and carbon signals are quoted relatively to tetramethylsilane (TMS) as an internal standard and expressed in parts per million (ppm). Electrospray ionization (ESI) mass spectrometry (MS) spectra were recorded using a Waters Micromass Q-Tof micro mass instrument, while high resolution (HR)-ESI mass spectra were recorded on an FT-ICR APEX_II_ spectrometer (Bruker Daltonics, Billerica, MA, USA).

#### 5.1.2. Synthesis of Mono Dithio-Dibutyroate Squalenoyl-hexaTMS-Trehalose **4a** and Bis Dithio-Dibutyroate Squalenoyl-hexaTMS-Trehalose **4b**

To a stirred solution of hexaTMS-protected trehalose 2 (317 mg, 0.409 mmol) in dry toluene (12.7 mL) at rt under nitrogen atmosphere, EDC^.^HCl (78.5 mg, 0.409 mmol) was added, followed by DMAP (5.0 mg, 0.0409 mmol). The carboxysqualene-linker adduct **3** (298 mg, 0.491 mmol) was added after 30 min, the reaction mixture was heated at 50 °C and further stirred for 46 h. Reaction completion was confirmed via TLC (eluent mixture: 9:1 n-hexane/AcOEt). Then, the solvent was removed at reduced pressure, and the resulting crude oil was purified by direct silica gel flash chromatography (eluent mixture: 9:1 n-hexane/AcOEt) to provide pure **4a** (168 mg, 0.123 mmol, 30% yield) and pure **4b** (112 mg, 0.0573 mmol, 14% yield) as two colorless oils.


*Analytical characterization*



**4a:**


**^1^H-NMR** (CDCl_3_, 400 MHz): δ(ppm) = 5.19–5.05 (m, 5H), 4.93–4.89 (m, 2H), 4.31 (dd, *J* = 11.8, 2.1 Hz, 1H), 4.15–3.95 (m, 4H), 3.93–3.78 (m, 3H), 3.71–3.66 (m, 2H), 3.53–3.38 (m, 4H), 2.73–2.70 (m, 4H), 2.52–2.41 (m, 4H) 2.11–1.96 (m, 18H), 1.75–1.69 (m, 2H), 1.68 (s, 3H), 1.62–1.57 (m, 15H), 1.29–1.22 (m, 4H), 0.34–0.07 (m, 54H).

**^13^C-NMR** (CDCl_3_, 101 MHz): δ(ppm) 173.0, 172.9, 135.1, 135.0, 134.9, 133.6, 131.3, 125.1, 124.41, 124.38, 124.29, 124.27, 94.5, 94.4, 73.4, 73.3, 73.0, 72.8, 72.6, 71.9, 71.4, 70.7, 64.3, 63.5, 61.7, 39.76, 39.73, 39.68, 37.8, 37.6, 35.8, 32.6, 32.4, 29.7, 28.3, 26.9, 26.8, 26.68, 26.67, 25.7, 24.2, 24.0, 17.7, 16.06, 16.05, 16.01, 15.9, (1.0, 0.90, 0.86, 0.2, 0.1 = 18C).

**MS (ESI^+^)**, *m*/*z*: calcd for C_65_H_126_O_14_S_2_Si_6_ 1362.72, found 1385.71 (M + Na^+^).


**4b:**


**^1^H-NMR** (CDCl_3_, 400 MHz): δ(ppm) = 5.21–5.08 (m, 10H), 4.96–4.91 (m, 2H), 4.34–4.27 (m, 2H), 4.13–3.97 (m, 8H), 3.92 (t, *J* = 8.9 Hz, 2H), 3.53–3.45 (m, 4H), 2.78–2.70 (m, 8H), 2.56–2.44 (m, 8H), 2.15–1.95 (m, 36H), 1.80–1.72 (m, 4H), 1.71 (s, 6H), 1.62 (s, 30H), 1.33–1.26 (m, 8H), 0.21–0.08 (m, 54H).

**^13^C-NMR** (CDCl_3_, 101 MHz): δ(ppm) = 173.8 (2C), 173.6 (2C), 135.0 (2C), 134.9 (2C), 134.8 (2C), 133.6 (2C), 131.2 (2C), 125.1 (2C), 124.4 (4C), 124.3 (4C), 94.4 (2C), 73.5 (2C), 72.7 (2C), 71.9 (2C), 70.7 (2C), 63.9 (2C), 63.3 (2C), 39.7 (4C), 35.8 (2C), 34.3 (2C), 34.1 (2C), 29.7 (2C), 29.1 (6C), 28.8 (2C), 28.2 (4C), 26.9 (2C), 26.8 (2C), 26.6 (4C), 25.7 (4C), 25.0 (2C), 24.7 (2C), 17.7 (2C), 16.0 (4C), 15.8 (2C), (1.0, 0.9, 0.4, 0.2 = 18C).

**MS (ESI^+^)**, *m*/*z*: calcd for C_100_H_182_O_17_S_4_Si_6_ 1951.09, found 1974.13 (M + Na^+^).

#### 5.1.3. Synthesis of Mono Dithio-Dibutyroate Squalenoyl-Trehalose **1a**

To a stirred solution of **4a** (120 mg, 0.0880 mmol) in MeOH (1.3 mL) at rt, acetic acid (5.3 μL, 0.0880 mmol) was added by syringe; the resulting reaction mixture was warmed to 40 °C and left under stirring overnight. Reaction completion was confirmed via TLC (eluent mixture: 9:1 n-hexane/AcOEt) so that the solvent was removed at reduced pressure. The resulting crude solid was purified by direct silica gel flash chromatography (eluent mixture: 85:15 CH_2_Cl_2_/MeOH) to yield pure **1a** (80.2 mg, 0.0854 mmol, 90% yield) as a white solid.


*Analytical characterization*


**^1^H NMR** (DMSO-*d*_6_, 400 MHz): δ(ppm) = 5.17–5.03 (m, 6H), 4.88 (dd, *J* = 3.9, 2.4 Hz, 2H), 4.85 (t, *J* = 3.9 Hz, 1H), 4.77 (dd, *J* = 5.0, 2.7 Hz, 2H), 4.68 (dd, *J* = 6.1, 2.2 Hz, 2H), 4.34 (t, *J* = 5.9 Hz, 1H), 4.29–4.21 (m, 1H), 4.10–4.02 (m, 1H), 3.97 (t, *J* = 6.6 Hz, 2H), 3.95–3.88 (m, 1H), 3.69–3.61 (m, 1H), 3.57–3.54 (m, 3H), 3.51–3.43 (m, 1H), 3.30–3.21 (m, 2H), 3.19–3.08 (m, 2H), 2.76–2.66 (m, 5H), 2.41 (q, *J* = 6.9 Hz, 4H), 2.07–1.83 (m, 22H), 1.70–1.60 (m, 5H), 1.56 (s, 15H).

**^13^C-NMR** (DMSO-*d*_6_, 101 MHz): δ(ppm): 172.4, 172.2, 134.43, 134.38, 134.31, 133.6 (2C), 124.4, 124.14, 124.08 (2C), 123.9, 98.22, 98.15, 72.84, 72.77, 72.61, 71.57, 71.47, 70.1 (2C), 69.6, 63.5, 63.3, 60.8, 39.4, 39.2, 36.7, 36.6, 35.3, 32.0 (2C), 27.8 (2C), 26.3, 26.2, 26.0 (2C), 25.5, 24.0, 23.9, 17.6, 15.8 (4C), 15.7.

**MS (ESI^+^)**, *m*/*z*: calcd for C_47_H_78_O_14_S_2_ 930.48, found 953.51 (M + Na^+^).

#### 5.1.4. Synthesis of Bis Dithio-Dibutyroate Squalenoyl-Trehalose **1b**

To a stirred solution of **4b** (65.2 mg, 0.0334 mmol) in MeOH (1 mL) at rt, acetic acid (1.9 μL, 0.0334 mmol) was added by syringe, the reaction mixture was warmed and stirred at 40 °C overnight. After checking reaction completion by TLC (eluent mixture: 9:1 n-hexane/AcOEt), the solvent was removed at reduced pressure, and a crude residue was purified by direct silica gel flash chromatography (eluent mixture: 85:15 CH_2_Cl_2_/MeOH) to provide pure target **1b** (51.2 mg, 0.0310 mmol, 94% yield) as a white solid.


*Analytical characterization*


**^1^H-NMR** (DMSO-*d*_6_, 400 MHz): δ(ppm) = 5.15–5.00 (m, 12H), 4.95–4.85 (m, 2H), 4.83 (d, *J* = 3.5 Hz, 2H), 4.80–4.73 (m, 2H), 4.26 (d, *J* = 10.4 Hz, 2H), 4.09–4.02 (m, 2H), 3.96 (t, *J* = 6.6 Hz, 4H), 3.94–3.86 (m, 2H), 3.62–3.49 (m, 4H), 3.32–3.23 (m, 4H), 3.17–3.08 (m, 2H), 2.71 (td, *J* = 7.4, 2.7 Hz, 8H), 2.40 (q, *J* = 7.0 Hz, 8H), 1.95 (ddq, *J* = 37.7, 21.4, 6.9 Hz, 44H), 1.68–1.61 (m, 10H), 1.55 (s, 30H).

**^13^C-NMR** (DMSO-*d*_6_, 101 MHz): δ(ppm): 172.7 (2C), 172.6 (2C), 134.85 (2C), 134.80 (2C), 134.74 (2C), 134.0 (2C), 131.1 (2C), 124.9 (2C), 124.6 (2C), 124.51 (2C), 124.47 (2C), 124.36 (2C), 94.1 (2C), 73.2 (2C), 71.9 (2C), 70.5 (2C), 70.2 (2C), 63.9 (2C), 63.7 (2C), 39.7 (2C), 39.6 (2C), 37.2 (2C), 37.1 (2C), 35.7 (2C), 32.5 (4C), 29.1 (4C), 28.2 (4C), 26.9 (2C), 26.8 (2C), 26.7 (2C), 26.4 (2C), 25.9 (2C), 24.9 (2C), 24.4 (2C), 18.0 (2C), 16.4 (2C), 16.2 (2C), 16.1 (2C).

**MS (ESI^+^)**, *m*/*z*: calcd for C_82_H_134_O_17_S_4_ 1518.85, found 1541.84 (M + Na^+^).


*NA assembly and characterization*


**Mono—****Sq-NA 5a**. The squalene-trehalose conjugate **1a** (4.0 mg) was dissolved in THF (1 mL), the clear solution was slowly added to magnetically stirred (500 rpm) MilliQ grade distilled water (2 mL) in a round bottom flask, and the resulting solution was stirred for 5 additional min. The solvent was then thoroughly evaporated at reduced pressure, obtaining pure **mono—Sq-NA 5a** as an opalescent suspension (2 mL, 2 mg/mL), in accordance with standard solvent evaporation protocols [32]. 

**Bis—Sq-NA 5b**. The same procedure for **Mono—Sq-NA 5a** was followed, obtaining pure **bis—Sq-NA 5b** as opalescent suspension (2 mL, 2 mg/mL).

**NA Characterization**. A 90 Plus Particle Size Analyzer (Brookhaven Instrument Corporation, Holtsville, NY, USA) was used to determine the hydrodynamic diameter of our NAs by dynamic light scattering (DLS), operating a solid-state laser (λ = 661 nm) at 15 mW, and with a scattering angle of 90 °C. The same instrument equipped with an AQ-809 electrode, operating at an applied voltage of 120 V at 25 °C, was used to determine the ζ-potential of our NAs.

In detail, each NA sample was prepared first by diluting it at 0.6 mg/mL and by sonication for 3 min. Then, each sample was analyzed in ten independent measurements lasting 60 s each. The hydrodynamic diameters of our NAs were calculated using the Mie theory, assuming refractive index values and absolute viscosity of the medium to be, respectively, 0.890 cP and 1.33. Their Z-potentials were extrapolated using the Smoluchowski theory, from the electrophoretic mobility of our NAs.

### 5.2. Cell Cultures

We cultured HeLa cells (ATCC: CCL-2) in DMEM (Euroclone, Pero, Italy) medium, supplemented with 10% fetal bovine serum (FBS; Euroclone), 1% penicillin/streptomycin (Gibco, 15140122) and 1% L- glutamine (Gibco, Grand Island, NV, USA) in a humidified atmosphere of 5% CO_2_ at 37 °C. Cells were passaged one or two times a week depending on confluency using Trypsin-EDTA 0.05% (Gibco, 25300054).

### 5.3. Cytotoxicity Assay

To measure culture viability, we performed the 3-(4,5-dimethylthiazol-2-yl)-2,5-diphenyltetrazolium bromide (MTT) assay. HeLa cells were cultured in a 96-well plate at a concentration of 5 × 10^3^ cell/cm^2^ and incubated at 37 °C for 24 h. Then, the cell medium was replaced with MTT diluted in PBS at a final concentration of 0.25 mg/mL for 30 min at 37 °C. Afterwards, **the** MTT solution was carefully removed, and formazan precipitates collected in 200 μL of DMSO. The absorbance measured at 570 nm using a spectrophotometer reflects cell vitality. Relative cell vitality was expressed as fold over control, set at 100%.

### 5.4. Autophagy Assay

We monitored autophagy by Western blotting as previously reported [19]. First, cells were washed in PBS and lysed in RIPA buffer (150 mM NaCl, 50 mM HEPES, 0.5% NP40, 1% sodium-deoxycholate) containing protease inhibitors (Merck). After collecting the material, samples were briefly vortexed three times every 10 min and then any solid was removed by centrifugation at 10,000× *g* for 10 min. We performed all experimental procedures at 4 °C. The determination of lysates’ protein concentrations was carried out via Bradford assay (Bio-Rad Laboratories, Segrate, Italy). For Western blotting experiments, an equal amount of proteins was diluted with 0.25% Laemmli buffer and boiled at 95 °C for 10 min. We ran protein samples onto 15% SDS-PAGE gels. After gel electrophoresis, the proteins were transferred onto a polyvinylidene difluoride (PVDF) membrane (Sigma-Aldrich, St. Louis, MO, USA) at 90 V for 90 min at 4 °C. Used primary antibodies (source in parentheses) include mouse anti-LC3B, 1:500 (Enzo LifeScience, New York, NY, USA) and mouse anti α-tubulin 1:5000 (Sigma Aldrich), which were applied overnight in blocking buffer (20 mM Tris, pH 7.4, 150 mM NaCl, 0.1% Tween 20, and 5% nonfat dry milk). We detected proteins by the ECL prime detection system (GE Healthcare, Chicago, IL, USA). We acquired an ECL signal via the imaging ChemiDoc Touch Imaging system Version 2.1.0.35 (Bio-Rad Laboratories). We measured the optical density of the specific bands with ImageLab software Version 6.0.1 build 34 (2017, Bio-Rad Laboratories, Inc.).

### 5.5. Autophagosome–Lysosome Fusion Assay

The autophagosome–lysosome fusion assay was performed in HeLa cells transfected with the mCherry- and GFP-tagged LC3B [23]. using PEI (jetPEI^®^ reagent). One day after transfection, we treated cells with trehalose (100 mM), **5a** (25 μM) and **5b** (10 μM) for 24 h. At the end of the treatment, we fixed cells with 4% paraformaldehyde (10 min, rt). We mounted the cover slips with prolonged reagent (Life Technologies, Carlsbad, CA, USA). Images were acquired with the optical microscope ZEISS AXIO IMAGER M2 using a plan-Apochromat 40× objective and 0.102 μm × 0.102 μm pixel size. We counted the numbers of mCherry and/or GFP-positive puncta in each cell using ImageJ.

### 5.6. Statistical Analysis

All data are mean ± standard error of the mean (SE). We exploited GraphPad Prism to analyze our entire data set. We applied unpaired Student’s *t*-test (two classes) or ANOVA followed by Tukey’s post hoc test (more than two classes). The number of experiments (*n*) and level of significance (*p*) were indicated throughout the text.

## Data Availability

Data are available contacting the corresponding authors upon reasonable request.

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
