# Peer review of "Squalene-Based Nano-Assemblies Improve the Pro-Autophagic Activity of Trehalose"

_pharmaceutics, 2022, doi:10.3390/pharmaceutics14040862_

Round 1

Reviewer 1 Report

Measuring the apparent amount of LC3-II on a blot alone does not predict absolute autophagic activity.
The increase in LC3-II merely indicates the accumulation of autophagosomes and does not guarantee autophagic degradation. However, if the amount of LC3-II further accumulates in the presence of a lysosomal protease inhibitor, this would indicate the promotion of autophagic flux. If, however, the amount of LC3-II remains unchanged, then the accumulation of autophagic flux is probably due to inhibition of autophagic degradation, such as inhibition of autophagosome/lysosome fusion.

Author Response

We are glad that the reviewers found our manuscript of interest. Their criticisms were treasured. We are resubmitting a fully improved version of the manuscript that addresses the Reviewers’ concerns and includes a total of 6 new result panels.

We believe that the current manuscript version is strongly improved with the inclusion of the experiments highlighted above, and we hope that it is now suitable for publication in Pharmaceutics.

Sincerely,

Giovanni Piccoli

Reviewer 1

Measuring the apparent amount of LC3-II on a blot alone does not predict absolute autophagic activity.
The increase in LC3-II merely indicates the accumulation of autophagosomes and does not guarantee autophagic degradation. However, if the amount of LC3-II further accumulates in the presence of a lysosomal protease inhibitor, this would indicate the promotion of autophagic flux. If, however, the amount of LC3-II remains unchanged, then the accumulation of autophagic flux is probably due to inhibition of autophagic degradation, such as inhibition of autophagosome/lysosome fusion.

A: we performed the experiment suggested by the Reviewer. In detail, upon 22 hours treatment with the NAs, we added bafilomycin 100 nM for 2 hours. Then, we analyzed autophagy by western-blotting. These new experiments suggested that NA 5b is a bona fide autophagic inducer and has stronger activity than 5a. Indeed, 5a and 5b showed a similar activity as revealed by our image-based assessment of the autophagic flux (figure 5). The lack of an overt effect of 5a once co-administered with bafilomycin may be due to specific experimental conditions. We may need to optimize timing and concentration. The data are presented in new figure 4A-B.

Reviewer 2 Report

The paper is well organized. It can be accepted in current form.

Author Response

We are glad that the reviewers found our manuscript of interest. Their criticisms were treasured. We are resubmitting a fully improved version of the manuscript that addresses the Reviewers’ concerns and includes a total of 6 new result panels.

We believe that the current manuscript version is strongly improved with the inclusion of the experiments highlighted above, and we hope that it is now suitable for publication in Pharmaceutics.

Sincerely,

Giovanni Piccoli

Reviewer 2

The paper is well organized. It can be accepted in current form.

A: we thank this reviewer for his/her support.

Reviewer 3 Report

The paper submitted by Frapporti et al. deals with the synthesis and characterization of some new trehalose-based derivatives in order to increase it therapeutic effects.

The manuscript is clear and the conclusions are supported by the results. However, some corrections are necessary in order to increase the overall quality of the paper:

  1. the abstract section is too general. the authors must indicate specific results as the size, ZP values etc
  2. the authors must explain why the sample 5a has the same hydrodynamic diameter as sample 5b?! normally, if the molecular weight of a molecule increases, the HD increases also...
  3. the discussion section is quite general also. the authors have not taken into discussion their specific results. 
  4. an conclusion section must be inserted in the manuscript. 

Author Response

We are glad that the reviewers found our manuscript of interest. Their criticisms were treasured. We are resubmitting a fully improved version of the manuscript that addresses the Reviewers’ concerns and includes a total of 6 new result panels.

We believe that the current manuscript version is strongly improved with the inclusion of the experiments highlighted above, and we hope that it is now suitable for publication in Pharmaceutics.

Sincerely,

Giovanni Piccoli

Reviewer 3

The paper submitted by Frapporti et al. deals with the synthesis and characterization of some new trehalose-based derivatives in order to increase it therapeutic effects.

The manuscript is clear and the conclusions are supported by the results. However, some corrections are necessary in order to increase the overall quality of the paper:

  1. the abstract section is too general. the authors must indicate specific results as the size, ZP values etc

A: we included relevant information in the abstract

  1. the authors must explain why the sample 5a has the same hydrodynamic diameter as sample 5b?! normally, if the molecular weight of a molecule increases, the HD increases also...

A: we thank the reviewer for this comment. Generally, an increment of the molecular weight may correspond to an increase of the size of the nanoassemblies. However, in our case, as proposed in the manuscript (see scheme 2), we believe that the two residues of squalene present in compound 5b self-assemble in order to display on the outer surface the trehalose residues. Therefore, the resulting bis-squalenylated nanoassemblies 5b have a more hydrophobic core and do not increase their HD in respect to monosqualenylated 5a.

  1. the discussion section is quite general also. the authors have not taken into discussion their specific results. 

A: we did our best to improve the Discussion, and we believe it now to be of good quality.

  1. an conclusion section must be inserted in the manuscript. 

A: we modified the text accordingly.

Reviewer 4 Report

In this manuscript, Frapporti et al. present an evaluation of in vitro autophagy stimulation in HeLa cells by squalene-based trehalose nano assemblies (NAs). This research merits interest, as trehalose is indeed a possible potent and safe inducer of autophagy, but in vivo administration requires high doses with low efficacy. Modifications that may increase the in vivo potency of trehalose are therefore very interesting.

However, the findings in this manuscript that these NAs are potent autophagy inducers are not convincing. I have several major remarks regarding the experiments, conclusions and presentation of the manuscript:

  1. The title and further conclusions in the manuscript mention that squalene-based NAs IMPROVE the autophagic effect of trehalose. However, the Western blots clearly show higher autophagy levels in the trehalose-treated compared to the NAs-treated cells. Of course, a higher dose of trehalose was evaluated, but no comparison was made with similar doses of trehalose and NAs 5a and 5b. Therefore, the statement in the title and the conclusions that NAs of trehalose improve the effect of trehalose alone cannot be derived from these data. However, such a comparison is nonetheless very important as, based on the Western blots, the effect of NAs 5a and 5b at non-toxic doses seems very limited.
  2. The LC3 Western blot experiments should be repeated in absence and presence of lysosomal inhibitors to make any assumptions regarding the autophagic flux. The GFP-mCherry-LC3B assay alone is not adequate, especially since there are issues regarding their presentation/quantification (see further). In addition, additional markers (such as Sqstm1/p62) could be assessed.
  3. Please revise all figure legends. Figure legends should accompany figure so that figures can already be understood without the text in the manuscript. Many abbreviations used in the figures are not explained in the legends. In Figure 2, concentration of trehalose is not mentioned.
  4. The MTT assay (Fig. 1) measures metabolic activity, and not cell viability directly. Are the treatments that lead to a reduction in the MTT assay accompanied by an increase in cell death?
  5. The Western blot in Figure 2B is obviously cut. Please show the full blot to make sure that the same exposure was used to quantify the bands.
  6. It is unclear why 25 µM of 5b was assessed on Western blotting (Fig. 2-3), if it is cytotoxic. This seems irrelevant (enhanced autophagy is expected in stressed/dying cells).
  7. The significant increase in Fig. 3A of 10uM 1b and 10 uM 5a is not observed from the Western blot in Fig. 2A and C.
  8. LC3BII/LC3BI is quantified in Fig. 3B. However, LC3BI seems barely detectable in the Western blots in Fig. 2. If this band intensity is near the background, quantification is prone to errors. Please show enhanced exposure of the blots in Fig. 2 to better assess the LC3BI band.
  9. A similar remark can be made about LC3BII bands, for example in Fig. 2B (control) and Fig. 6C (control). Comparison with a band intensity so close to background will lead to overassumptions (such as ca. 20-fold increases).
  10. The relevance of Fig. 4 is unclear. It represents dose responses with only 3 or 4 concentrations, and the curves are not sigmoidal, so the accuracy of the calculated EC50s is doubtful. The figure therefore does not represent any useful data that was not presented in Fig. 2 and 3.
  11. The merged images in Fig. 5 show only yellow dots. Nonetheless, this should show yellow (autophagosomes) and red (autolysosomes) dots. Also, the quantification shows that many dots should be autolysosomal (and therefore only red). How can this be explained?
  12. The comparison between Fig. 5B and C is confusing. In Fig. 5C, ca. 70% is autophagosome and ca. 30% autolysosome. However, in Fig. 5B, for example in control conditions, ca. 40 out of 100 are autophagosome, corresponding to 40%. How does this lead to 70% in Fig. 5C?
  13. The quantifications in Fig. 5 also shows an average of ca. 100 dots per cell. This high number is not apparent in the representative images.
  14. The quantifications in Fig. 7 are not observed in the Western blots of Fig. 6. In Fig. 6C, all LC3BII bands are higher than the control LC3BII band, but this is not so in the quantification. In addition, the lower LC3BII levels in 5b+BSO compared to 5b is not observed in the Western blot of Fig. 6C (there LC3BII even seem higher in 5b+BSO). In Fig. 6D, LC3BII seems lower or similar in 5a compared to control (while the quantification shows an increase.
  15. Figure 2B-C is referred to in the text for the effects of 1b. This should be Figure 2A.
  16. In Figure 3B, the label on the X-axis reads 5b 15uM. This should be 25uM.

Author Response

We are glad that the reviewers found our manuscript of interest. Their criticisms were treasured. We are resubmitting a fully improved version of the manuscript that addresses the Reviewers’ concerns and includes a total of 6 new result panels.

We believe that the current manuscript version is strongly improved with the inclusion of the experiments highlighted above, and we hope that it is now suitable for publication in Pharmaceutics.

Sincerely,

Giovanni Piccoli

Reviewer 4

In this manuscript, Frapporti et al. present an evaluation of in vitro autophagy stimulation in HeLa cells by squalene-based trehalose nano assemblies (NAs). This research merits interest, as trehalose is indeed a possible potent and safe inducer of autophagy, but in vivo administration requires high doses with low efficacy. Modifications that may increase the in vivo potency of trehalose are therefore very interesting.

However, the findings in this manuscript that these NAs are potent autophagy inducers are not convincing. I have several major remarks regarding the experiments, conclusions and presentation of the manuscript:

  1. The title and further conclusions in the manuscript mention that squalene-based NAs IMPROVE the autophagic effect of trehalose. However, the Western blots clearly show higher autophagy levels in the trehalose-treated compared to the NAs-treated cells. Of course, a higher dose of trehalose was evaluated, but no comparison was made with similar doses of trehalose and NAs 5a and 5b. Therefore, the statement in the title and the conclusions that NAs of trehalose improve the effect of trehalose alone cannot be derived from these data. However, such a comparison is nonetheless very important as, based on the Western blots, the effect of NAs 5a and 5b at non-toxic doses seems very limited.

A: we analyzed autophagy in HeLa cells upon treatment with trehalose, 5a, and 5b at 25 M. These new results suggest that the NAs are more efficient than trehalose in inducing autophagy. The data are presented in new figure 4A-B.

  1. The LC3 Western blot experiments should be repeated in absence and presence of lysosomal inhibitors to make any assumptions regarding the autophagic flux. The GFP-mCherry-LC3B assay alone is not adequate, especially since there are issues regarding their presentation/quantification (see further). In addition, additional markers (such as Sqstm1/p62) could be assessed.

A: we performed the experiment suggested by the Reviewer. In detail, upon 22 hours treatment with the NA, we added bafilomycin 100 nM for 2 hours. Then, we analyzed autophagy by western-blotting. These new experiments suggested that NA 5b is a bona fide autophagic inducer and has stronger activity than 5a. The data are presented in new figure 4A-B.

  1. Please revise all figure legends. Figure legends should accompany figure so that figures can already be understood without the text in the manuscript. Many abbreviations used in the figures are not explained in the legends. In Figure 2, concentration of trehalose is not mentioned.

A: We revised all figure legends

  1. The MTT assay (Fig. 1) measures metabolic activity, and not cell viability directly. Are the treatments that lead to a reduction in the MTT assay accompanied by an increase in cell death?

A: MTT is a well accepted assay to measure cell viability. We may suggest some reviews on this topic (PMID: 29496266; 34307082). We have exploited MTT to measure cell viability in several former publications (PMID: 33876242; 29541021; 31434235).

  1. The Western blot in Figure 2B is obviously cut. Please show the full blot to make sure that the same exposure was used to quantify the bands.

A: Full blots at low and higher exposure are included as supplementary material.

  1. It is unclear why 25 µM of 5b was assessed on Western blotting (Fig. 2-3), if it is cytotoxic. This seems irrelevant (enhanced autophagy is expected in stressed/dying cells).

A: Initially, we compared 5a and 5b at different concentrations, the aim being to identify the safer and more potent compound.

  1. The significant increase in Fig. 3A of 10uM 1b and 10 uM 5a is not observed from the Western blot in Fig. 2A and C.

A: we run additional experiments and provided more representative blots. Data are included in new figure

  1. LC3BII/LC3BI is quantified in Fig. 3B. However, LC3BI seems barely detectable in the Western blots in Fig. 2. If this band intensity is near the background, quantification is prone to errors. Please show enhanced exposure of the blots in Fig. 2 to better assess the LC3BI band.

A: Full blots at low and higher exposure are included as supplementary material.

  1. A similar remark can be made about LC3BII bands, for example in Fig. 2B (control) and Fig. 6C (control). Comparison with a band intensity so close to background will lead to overassumptions (such as ca. 20-fold increases).

A: Full blots at low and higher exposure are included as supplementary material. We included a new representative blot (figure 6C).

  1. The relevance of Fig. 4 is unclear. It represents dose responses with only 3 or 4 concentrations, and the curves are not sigmoidal, so the accuracy of the calculated EC50s is doubtful. The figure therefore does not represent any useful data that was not presented in Fig. 2 and 3.

A: following the reviewer’s suggestion, we removed that figure in this new version.

  1. The merged images in Fig. 5 show only yellow dots. Nonetheless, this should show yellow (autophagosomes) and red (autolysosomes) dots. Also, the quantification shows that many dots should be autolysosomal (and therefore only red). How can this be explained?

A: we reconsidered our analysis and set more stringent parameters for object recognition. We also provided more representative images. Data are included in new figure 5.

  1. The comparison between Fig. 5B and C is confusing. In Fig. 5C, ca. 70% is autophagosome and ca. 30% autolysosome. However, in Fig. 5B, for example in control conditions, ca. 40 out of 100 are autophagosome, corresponding to 40%. How does this lead to 70% in Fig. 5C?

A: we reconsidered our analysis and set more stringent parameters for the object recognition. Data are included in new figure 5.

  1. The quantifications in Fig. 5 also shows an average of ca. 100 dots per cell. This high number is not apparent in the representative images.

A: we reconsidered our analysis and set more stringent parameters for the object recognition. Data are included in new figure 5.

  1. The quantifications in Fig. 7 are not observed in the Western blots of Fig. 6. In Fig. 6C, all LC3BII bands are higher than the control LC3BII band, but this is not so in the quantification. In addition, the lower LC3BII levels in 5b+BSO compared to 5b is not observed in the Western blot of Fig. 6C (there LC3BII even seem higher in 5b+BSO). In Fig. 6D, LC3BII seems lower or similar in 5a compared to control (while the quantification shows an increase.

A: we run additional experiments and provided more representative blots. Data are included in new figure 6.

  1. Figure 2B-C is referred to in the text for the effects of 1b. This should be Figure 2A.

A: we are sorry for the mistake, and we corrected the text accordingly to the reviewer’s request.

  1. In Figure 3B, the label on the X-axis reads 5b 15uM. This should be 25uM.

A: we are sorry for the mistake and we corrected the graph accordingly to the reviewer’s request.

Round 2

Reviewer 4 Report

The authors have clearly put effort in improving the manuscript. The figure with Bafilomycin is now an important asset showing that 5b is clearly the most potent NA for stimulation of the autophagic flux. However, I still have an issue with the representative images in Fig. 5, as I see no red punctae, while the quantification clearly shows there should be some. Also the control image only shows 5 quite large yellow dots. Other representative images are therefore recommended.

Author Response

We included new images acquired with longer exposure time to enhance mCherry signal.  We acquired all images with the same parameters